# Influence of agriculture on child nutrition through child feeding practices in India: A district-level analysis

**Deepshikha Dey**[1], **Arup Jana**[1], **Manas Ranjan Pradhan**[2]*

**1** International Institute for Population Sciences (IIPS), Mumbai, Maharashtra, India, **2** Department of Fertility and Social Demography, International Institute for Population Sciences (IIPS), Mumbai, Maharashtra, India

* manasiips@gmail.com

**Data Availability Statement:** Secondary data available in public domain has been used for this paper. The NFHS-4 data is available at https://dhsprogram.com/Data/.The District-Wise Crop Production Statistics, 2015-16 published by the

## Abstract

Malnutrition continues to be a primary concern for researchers and policymakers in India. There is limited scientific research on the effect of agriculture on child nutrition in the country using a large representative sample. To the best of our knowledge, no study has examined the spatial clustering of child malnutrition and its linkage with agricultural production at the district-level in the country. The present study aims to examine agricultural production's role in improving the nutritional status of Indian children through child feeding practices. The nutritional indicators of children from the National Family Health Survey-4 (2015–16) and the agricultural production data for all the 640 districts of India obtained from the District-Wise Crop Production Statistics (2015–16), published by the Ministry of Agriculture, Government of India were used for the analysis. The statistical analysis was undertaken in STATA (version 14.1). ArcMap (version 10.3), and GeoDa (version 1.8) were used for the spatial analysis. The study found a higher prevalence of malnutrition among children who had not received Minimum Meal Frequency (MMF), Minimum Dietary Diversity (MDD), and Minimum Acceptable Diet (MAD). Further, child feeding practices- MMF, MDD, and MAD- were positively associated with high yield rates of spices and cereals. The yield rate of cash crops, on the contrary, harmed child feeding practices. Production of pulses had a significant positive effect on MDD and MAD. Districts with high cereal yield rates ensured that children receive MMF and MAD. There is a significant spatial association between child feeding practices and malnutrition across Indian districts. The study suggests that adopting nutrient-sensitive agriculture may be the best approach to improving children's nutritional status.

## Introduction

Malnutrition continues to be a primary concern for researchers and policymakers in India. India is home to the highest number of wasted children under five years of age and has been found to have made no progress in meeting the 2025 global nutrition target for wasting [1]. The Global Hunger Index, 2020, stated that India is suffering from a severe problem of hunger and malnutrition and ranks 94th out of 107 countries based on three leading indicators: proportion of undernourished people, under-five mortality rate, and prevalence of stunting and

Ministry of Agriculture, Government of India is available at https://eands.dacnet.nic.in/.

**Funding:** The author(s) received no specific funding for this work.

wasting among under-five children [2]. Nutrition is linked with agriculture in many ways [3], with agriculture being the primary source of food [4]. Agriculture plays a central role in nutrition by supplying nutritious, healthy, and affordable foods [5–7]. Agriculture is deemed to influence child feeding practices, which, in turn, are closely associated with child nutrition [8]. A geographical area with better dietary diversity is expected to have better feeding practices. Agriculture influences the feeding practices of an area and plays a significant role in the nutritional status of the area, given that the availability of different types of food crops influences dietary diversity [9]. Agricultural vulnerability to climate change was positively associated with malnutrition among Indian children [10]. Nutrition-sensitive agriculture interventions can improve children's diets [11]. Nutritionists recognize that one of the critical elements of high-quality diets is dietary diversity [12]. Dietary diversity acts as an intermediary variable between socioeconomic factors and nutritional outcomes [13]. World Health Organization (WHO) has observed that lack of appropriate dietary diversity and meal frequency among children aged six months or more, even with optimum breastfeeding, can result in malnourishment [14]. Thus, a nutritionally adequate diet is necessary for optimum growth, health, and development of children below two years. While dietary diversity is a proxy for adequate micronutrient density, minimum meal frequency is a proxy for a child's energy requirements [15, 16].

Feeding practices are crucial determinants of the nutritional status of infants and children [17]. Children's overall growth and development are affected by poor complementary feeding practices [18, 19]. Among children aged 6–23 months, a major determinant of chronic malnutrition is the lack of proper knowledge about infant and young child feeding (IYCF) and irregularities in household food distribution. The burden of childhood mortality can be largely reduced by improving IYCF practices [20]. The age group of fewer than two years is often referred to as the 'critical window' because inappropriate feeding practices can lead to chronic malnutrition in childhood. Stunting in early life has been found to have long-term effects on health, physical and cognitive development, and learning and earning potential [21–23]. Nutritious complementary foods, along with breastfeeding, can reduce stunting among children aged 6–23 months [24].

There has been limited scientific research on the effect of agriculture on child nutrition in India using a large representative sample. There is further a dearth of studies assessing this theoretically significant association between agricultural production and child nutrition at the district level. Agricultural production is expected to affect child nutritional status through availability of food items which influences child feeding practices. Additionally, the socio-demographic characteristics of the children, mother and household also conceptualized to influence child nutrition status (Fig 1). To the best of our knowledge, no study has examined the spatial clustering of child malnutrition and its linkage with agricultural production at the district level in the country. The present study examines agricultural production's role in improving the nutritional status of Indian children through child feeding practices. The study results will add to the existing evidence on the determinants of child malnutrition in India and formulation or strengthening strategies and programs for appropriate nutrition-focused interventions.

## Materials and methods

### Data

The study used nutritional data from the fourth round of the National Family Health Survey (NFHS), 2015–16. The NFHS-4 is a nationally representative survey of 601,509 households that provides information on various monitoring and impact evaluation indicators of health and nutrition. The survey's sampling design was a stratified two-stage one, with an overall

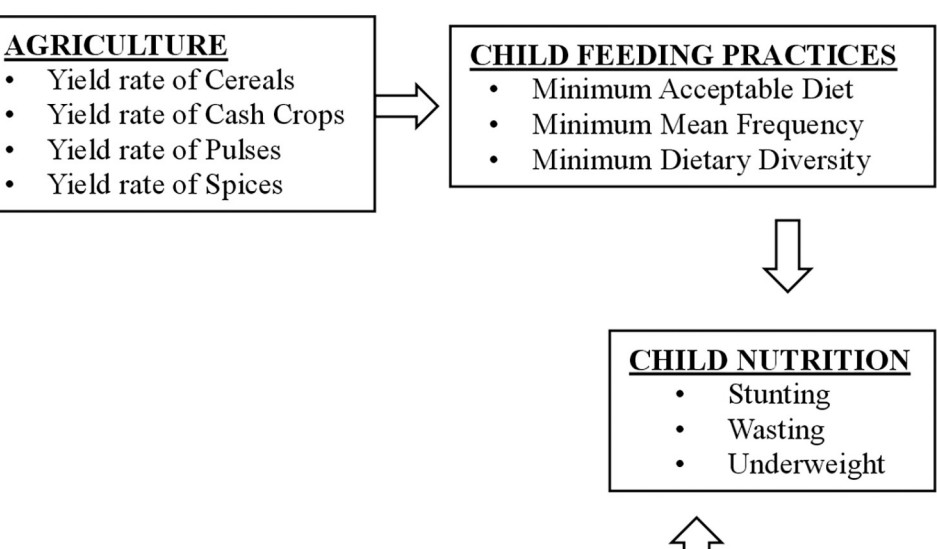

**Fig 1. Conceptual framework.**

response rate of 98%. The Primary Sampling Units (PSUs), which happened to be survey villages in rural areas and Census Enumeration Blocks (CEBs) in urban areas, were selected using probability proportional to size (PPS) sampling. Trained research investigators gathered the data using computer-assisted personal interviewing (CAPI). Only those respondents who gave voluntary consent were interviewed. For the present analysis, a sample of all the children aged 6–23 months who were living with their mothers and for whom information on the dietary practices was available (n = 71762) was considered. Additionally, agricultural production data for all the districts of India covered in NFHS-4 was obtained separately from the District-Wise Crop Production Statistics, 2015–16, published by the Ministry of Agriculture, Government of India [25].

## Study area

The nutritional outcome data used for the analysis represent all 640 districts across states of India that were covered in the NFHS-4. As per the Census 2011, there were 640 districts in

India, which was used as a sampling frame for the NFHS-4 survey, and thus, all those districts were covered in the survey. The agricultural production data were also gathered for all 640 districts. Therefore, the data used for this study is nationally representative.

## Dependent variables

The WHO IYCF indicators, namely Minimum Dietary Diversity (MDD), Minimum Meal Frequency (MMF), and Minimum Acceptable Diet (MAD), were used to assess the feeding practices among children aged 12–23 months. MDD is defined as the proportion of children aged 6–23 months receiving foods from four or more food groups. MMF is defined as the minimum number of times or more when a child, breastfed or non-breastfed, of age 6–23 months receives solid, semi-solid, or milk feeds. MAD is the proportion of children 6–23 months of age who receive a minimum acceptable diet (apart from breast milk). Stunting, wasting, and underweight were used to assess the nutritional status of the sample children. Stunting is defined as height-for-age, wasting is defined as weight-for-height, and underweight is defined as weight-for-age <-2 standard deviations (SD) of the WHO Child Growth Standards median respectively [26] Nutritional deprivation among children is measured by stunting, acute nutritional deficiency is gauged by wasting, and acute and chronic undernourishment is estimated by underweight.

## Independent variables

The average yield rate of production of different crops in the year 2015–16 was taken as the primary independent variable. This study conducted analysis at the disaggregated level by all districts and analyzed 42 crops grown all over India. The crops selected for the study were: (i) Cereals: rice, wheat, maize, jowar, ragi, bajra, small millets; (ii) Oilseeds: til, mustard, groundnut, castor, linseed, niger, sunflower, safflower; (iii) Vegetables: onion, potato, sweet potato; (iv) Spices: chilies, coriander, garlic, ginger, turmeric; (v) Cash Crops: sugarcane, jute, mesta, tobacco, cotton; and (vi) Pulses: kharif pulses, rabi pulses. Some socioeconomic and demographic variables were also considered in the analysis, including sex of child (male, female), age of the child in months (6–8, 9–11, 12–23), any incident of diarrhea during 15 days preceding the survey date (no, yes), low birth weight (no, yes), mother's education (illiterate, primary, secondary, higher), mother's body mass index (thin, normal, overweight, or obese), children ever born (1, 2, 3 & above), number of household members (<6, 6 & more), wealth index of household (poor, middle, rich), religion (Hindu, Muslim, others), caste (Scheduled Caste/Scheduled Tribe-SC/ST, Other Backward Classes-OBC, and others), and place of residence (rural, urban).

## Statistical analysis

Cross-tabulation was carried out to calculate the weighted percentage of the dependent and independent variables. Then, a chi-square test was employed to select the predictors. To assess the relationship between agriculture and IYCF indicators, an Ordinary Least Square (OLS) regression analysis was used. For the regression model, the IYCF variables were used as the dependent variables, and the average yield rate of crops was used as the independent variable after controlling for the sociodemographic characteristics.

An OLS regression was used with each outcome variable, that is, stunting, wasting, and underweight, to understand the association of malnutrition with its correlates. The independent variables were: proportion of age of the child, sex of the child, mother's education, BMI, MAD, MDD, MMF, diarrhea incidents, low birth weight, children ever born, religion, wealth status, caste, place of residence, and the number of household members. Univariate and

bivariate Local Moran's I statistics were used to present the auto-correlation between the dependent variables and the independent variables. To examine the geographical clustering of different variables used in the study, univariate LISA maps were employed. A bivariate LISA map was applied to measure the correlation between IYCF and child malnutrition status-stunting, wasting, and underweight.

The Spatial Error Model analysis was used to consider the effect of the variables that were absent in the regression model but had an effect on the outcome variables. A Spatial Error Model (SEM) is expressed in the following manner:

$$Yi = \beta Xj + \lambda + \varepsilon i$$

Here, Yi denotes the prevalence of malnutrition for the i[th] district, λ is the spatial autoregressive coefficient, Wij denotes the spatial weight of proximity between district i and j, Yj is the prevalence of malnutrition in the j[th] district, βj denotes the coefficient, Xj is the predictor variable, and εi is the residual. Maps were used to present the clustering of malnutrition, along with other determinants at the district level, in terms of hotspots and cold spots to help the planners and the policymakers plan and build new interventions for those specific underprivileged districts. The statistical analysis was undertaken in STATA (version 14.1). ArcMap (version10.3) and GeoDa (version1.8) were used for the spatial analysis.

## Results

Higher percentages of children without MMF, MDD, and MAD were found to be wasted and underweight than their counterparts (Table 1). For example- of the children without a MAD, 23% were wasted, and 34% were underweight. The corresponding figures were 20% and 30%, respectively, among those fed a MAD. Among children who were not fed with an MMF, 23% were wasted, and 34% were underweight, while it was 21% and 32% in children fed an MMF. Of the children without MDD, 23% were wasted and 35% underweight compared with 19% and 29% children with MDD. A higher percentage of boys, children from SC/ST groups, from rural areas, from poor wealth quintile, had illiterate/primary educated mothers, had mothers with 3+CEB, had thin mothers, were low birth weights, and aged 12–23 months were stunted than their respective counterparts. Wasting prevalence was higher among children from rural areas, children from SC/ST groups, from poor wealth quintile, had illiterate/primary educated mothers, had mothers with 3+CEB, had thin mothers and were low birth weights, had diarrhea in last two weeks preceding the survey, and aged <12 months than their respective counterparts. A higher percentage of boys, children from SC/ST groups, from rural areas, from poor wealth quintile, had illiterate/primary educated mothers, had mothers with 3+CEB, had thin mothers, were low birth weights, had diarrhea in the last two weeks preceding the survey, and aged 12–23 months were underweight than their respective counterparts.

The district-wise prevalence of stunting, wasting, and underweight is shown in Fig 2. Severe stunting was restricted to the northern part of the Indo-Gangetic Plain and the western Deccan plateau of the Indian subcontinent. By contrast, underweight and wasting were concentrated in the semi-arid and central highlands. A low prevalence of malnutrition was recorded in the southern and northeastern states/districts of India. The result of the OLS regression (Table 2 and Fig 3) shows the primary associations between the IYCF indicators–MAD, MDD, and MMF–and the average yield rate of crops after controlling for the sociodemographic factors. The regression result confirmed that the yield rate of cereals, cash crops, pulses, and spices across the districts closely determined MAD, MDD, and MMF. For MAD, the coefficient was the largest for cereals (coef. = 5.26, CI = 5.07, 5.44), followed by spices (coef. = 2.41, CI = 2.27, 2.54), and pulses (coef. = 2.28, CI = 2.06, 2.51) and represented a strong positive relationship.

**Table 1. Prevalence of malnutrition by background characteristics, India, 2015–16.**

| Determinants | Stunting | | Wasting | | Underweight | | Sample (n) |
|---|---|---|---|---|---|---|---|
| | < 2 SD | P Value | < 2 SD | P Value | < 2 SD | P Value | |
| **Minimum Acceptable Diet (MAD)** | | | | | | | |
| No | 36.24 | P<0.192 | 22.55 | p<0.001 | 33.95 | p<0.001 | 65,062(90.66) |
| Yes | 36.16 | | 19.92 | | 30.01 | | 6,700(9.34) |
| **Minimum Meal Frequency (MMF)** | | | | | | | |
| No | 36.22 | P<0.454 | 22.84 | p<0.001 | 34.44 | p<0.001 | 45,691(63.67) |
| Yes | 36.26 | | 21.33 | | 32.04 | | 26,071(36.33) |
| **Minimum Dietary Diversity (MDD)** | | | | | | | |
| No | 36.3 | P<0.137 | 23.24 | p<0.001 | 35.01 | p<0.001 | 55,901(77.9) |
| Yes | 36 | | 19.04 | | 28.54 | | 15,861(22.1) |
| **Religion** | | | | | | | |
| Hindu | 36.28 | p<0.001 | 22.79 | p<0.001 | 34.08 | p<0.001 | 51,826(72.22) |
| Muslim | 37.42 | | 21.05 | | 33.26 | | 11,333(15.79) |
| Others | 31.44 | | 18.81 | | 26.72 | | 8,603(11.99) |
| **Caste** | | | | | | | |
| Scheduled Caste/Scheduled Tribe | 40.77 | p<0.001 | 24.94 | p<0.001 | 39.66 | p<0.001 | 27,917(40.48) |
| OBC | 36.7 | | 22.03 | | 33.28 | | 28,180(40.86) |
| Others | 28.88 | | 19.28 | | 25.75 | | 12,864(18.65) |
| **Place of Residence** | | | | | | | |
| Urban | 30.68 | p<0.001 | 19.82 | p<0.001 | 27.05 | p<0.001 | 17,049(23.76) |
| Rural | 38.33 | | 23.29 | | 36.05 | | 54,713(76.24) |
| **Sex of the Child** | | | | | | | |
| Male | 38.39 | p<0.001 | 22.93 | p<0.001 | 35.63 | p<0.001 | 37,698(52.53) |
| Female | 33.89 | | 21.63 | | 31.38 | | 34,064(47.47) |
| **Wealth Index** | | | | | | | |
| Poor | 44.14 | p<0.001 | 26.32 | p<0.001 | 43.43 | p<0.001 | 34,832(48.54) |
| Middle | 34.59 | | 20.89 | | 30.96 | | 14,572(20.31) |
| Rich | 26.17 | | 17.69 | | 21.43 | | 22,358(31.16) |
| **Education Level** | | | | | | | |
| Illiterate/Primary | 45.47 | p<0.001 | 25.26 | p<0.001 | 43.48 | p<0.001 | 30,233(42.13) |
| Secondary | 31.85 | | 21.21 | | 29.12 | | 33,861(47.19) |
| Higher | 21.7 | | 16.57 | | 17.09 | | 7,668(10.69) |
| **Children Ever Born** | | | | | | | |
| 1 | 31.4 | p<0.001 | 21.65 | p<0.001 | 29.61 | p<0.001 | 26,072(36.33) |
| 2 | 35.47 | | 20.97 | | 31.42 | | 23,061(32.14) |
| 3 & more | 43.37 | | 24.73 | | 41.24 | | 22,629(31.53) |
| **Low birth weight** | | | | | | | |
| No | 31.92 | p<0.001 | 20.11 | p<0.001 | 28.02 | p<0.001 | 47,107(83) |
| Yes | 43.06 | | 28.78 | | 45.77 | | 9,648(17) |
| **Mother's Body mass Index** | | | | | | | |
| Thin | 41.44 | p<0.001 | 28.83 | p<0.001 | 44.82 | p<0.001 | 19,414(27.15) |
| Normal | 35.7 | | 21.19 | | 31.32 | | 43,005(60.14) |
| Overweight | 27.59 | | 15.96 | | 18.78 | | 6,577(9.2) |
| Obese | 25.6 | | 9.04 | | 18.23 | | 2,511(3.51) |
| **Had diarrhoea in last two weeks** | | | | | | | |
| No | 36.24 | P<0.813 | 21.78 | p<0.001 | 32.94 | p<0.001 | 61,482(85.67) |
| Yes | 36.24 | | 25.45 | | 37.36 | | 10,280(14.33) |

(*Continued*)

**Table 1.** (Continued)

| Determinants | Stunting | | Wasting | | Underweight | | Sample (n) |
|---|---|---|---|---|---|---|---|
| | < 2 SD | P Value | < 2 SD | P Value | < 2 SD | P Value | |
| **Number of Household Member** | | | | | | | |
| Less than 6 | 35.93 | P<0.527 | 22.17 | P<0.092 | 33.03 | p<0.005 | 42,403(59.09) |
| 6 & more | 36.69 | | 22.52 | | 34.43 | | 29,359(40.91) |
| **Age in Months** | | | | | | | |
| 6–8 | 20.48 | p<0.001 | 24.9 | p<0.001 | 27.13 | p<0.001 | 13,081(18.23) |
| 9–11 | 26.25 | | 25.1 | | 31.51 | | 12,118(16.89) |
| 12–23 | 43.11 | | 20.86 | | 35.9 | | 46,563(64.89) |

On the contrary, for cash crops (coef. = -0.75, CI = -0.77, -0.73), it represented a strong negative relationship. With respect to MDD, the coefficient was the highest for spices (coef. = 3.42, CI = 3.29, 3.54) and pulses (coef. = 1.86, CI = 1.66, 2.06), indicating a strong positive relationship and for cash crops (coef. = -0.05, CI = 0–0.07, -0.03), representing a strong negative relationship. For MMF, the coefficient was the high for cereals (coef. = 1.87, CI = 1.77, 1.97), spices (coef. = 1.48, CI = 1.41, 1.55), and pulses (coef. = 0.81, CI = 0.69, 0.93), representing a strong positive relationship and for cash crops (coef. = -0.32, CI = -0.33, -0.31), representing strong negative relationship.

The result depicting the extent of the spatial autocorrelation of malnutrition, IYFC indicators, and other socioeconomic determinants is presented by the univariate Local Moran's I statistics in **Table 3**. Autocorrelation is a characteristic of data that shows the degree of similarity between the values of the same variables over successive time intervals. The value of Moran's I was 0.770 for MDD, 0.301 for MAD, 0.442 for MMF, 0.193 for stunting, 0.591 for wasting, and 0.380 for underweight. The value of Moran's I was the highest for stunting, followed by wasting and underweight, depicting very clearly the geographical gradient of malnutrition in India. **Fig 4** presents the clustering of districts for MAD, MDD, and MMF. The districts of East Siang (Arunachal Pradesh), Baksa (Assam), Aizwal (Mizoram), Kurnool (Andhra Pradesh), Ahmedabad (Gujarat), and Hoshiarpur (Punjab) showed High-High clustering for MAD, making them hotspots. On the other hand, districts such as Kinnaur (Himachal Pradesh), Rampur

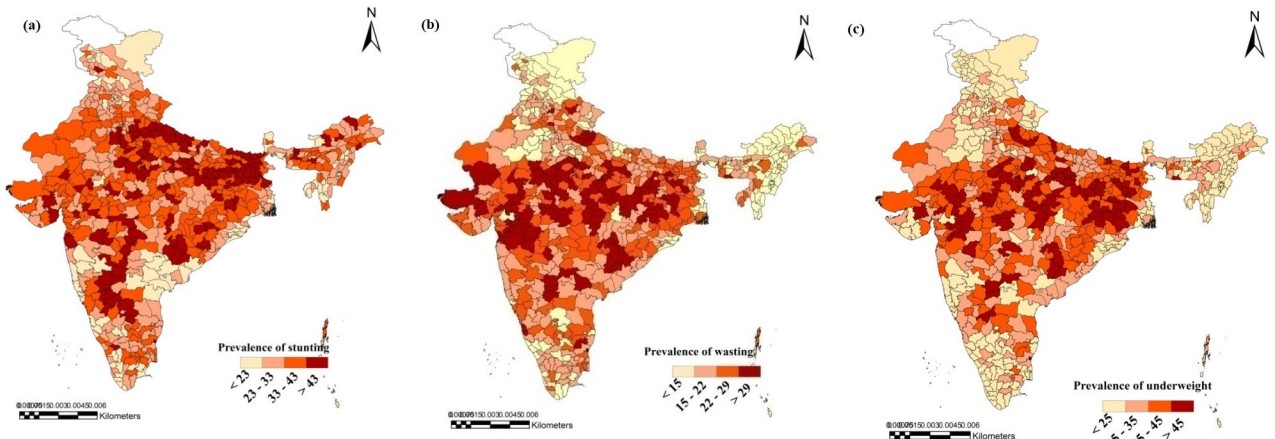

**Fig 2.** Prevalence of malnutrition a) Stunting, b) wasting, c) underweight for all districts, India, 2015–16.

**Table 2. Result of OLS regression analysis showing the relation between crop yield rate and MDD & MMF in India, 2015–16.**

| Determinants | Minimum Dietary Diversity (MDD) | Minimum Meal Frequency (MMF) |
|---|---|---|
| Average yield rate of cereal crops | 0.14 (-0.03,0.31) | 1.87 (1.77,1.97)*** |
| Average yield rate of cash crops | -0.05 (-0.07,-0.03)*** | -0.32 (-0.33,-0.31)*** |
| Average yield rate of pulses | 1.86 (1.66, 2.06)*** | 0.81 (0.69,0.93)*** |
| Average yield rate of spices | 3.42 (3.29, 3.54)*** | 1.48 (1.41,1.55)*** |
| Proportion of rich population | 1.20 (1.00, 1.40)*** | 0.98 (0.86, 1.11)*** |
| Proportion of Illiterate/primary educated mother | -1.85 (-2.19,-1.51)*** | -2.58 (-2.78,-2.38)*** |
| Proportion of non-Hindu | 0.80 (0.59,1.01)*** | 3.18 (3.06,3.30)*** |
| Proportion of urban population | 0.64 (0.41,0.86)*** | -0.15 (-0.29,-0.02)** |
| Proportion of general caste | 0.84 (0.60,1.09)*** | 0.63 (0.49,0.77)*** |
| Household member | -0.71 (-0.89,-0.54)*** | -1.24 (-1.34,-1.14)*** |
| Child ever born | 0.19 (-0.03,0.42) | -0.11 (-0.24,0.02) |

(Uttar Pradesh), Chandel (Manipur), Phek (Nagaland), and Jaintia Hills (Meghalaya) showed Low-Low clustering, making them cold spots. For MMF, High-High clustering was observed in the districts of Barmer (Rajasthan), Devbhumi Dwarka (Gujarat), and Bahraich (Uttar Pradesh), making them hotspots. In contrast, Low-Low clustering was observed in Kargil (Ladakh), Raigarh (Maharashtra), Kanniyakumari (Tamil Nadu), and West Garo Hills (Meghalaya), making them cold spots. As regards MDD, the figure reveals High-High clustering in the districts of Mahbubnagar (Telangana), Ratnagiri (Maharashtra), and Kanniyakumari(Tamil Nadu), making them hotspots, and Low-Low clustering in Kargil (Ladakh), Churu (Rajasthan), Ludhiana (Punjab), Nagpur (Maharashtra), and Almora (Uttarakhand), making them cold spots.

Table 4 exhibits the results of the bivariate Local Moran's I statistics for malnutrition against the socioeconomic correlates. MDD was found to have a negative spatial autocorrelation with underweight and wasting, with Moran's I values of 0.01 and 0.10, respectively. The

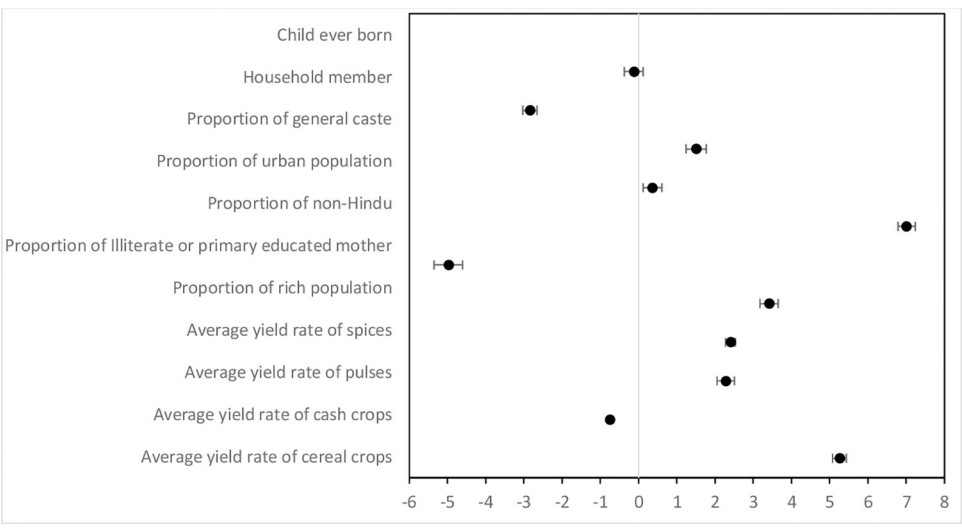

**Fig 3. Forest plot showing relation between crop yield rate and MAD in India, 2015–16.**

**Table 3. Univariate local Moran's I results for all districts (n = 640) among children aged 6–23 months, India, 2015–16.**

| Background Characteristics | Moran's I | P-Value |
| --- | --- | --- |
| Stunting | 0.193 | P<0.001 |
| Wasting | 0.591 | P<0.001 |
| Underweight | 0.380 | P<0.001 |
| Minimum Dietary Diversity | 0.770 | P<0.001 |
| Minimum Acceptable Diet | 0.301 | P<0.001 |
| Minimum Meal Frequency | 0.442 | P<0.001 |
| Percentage of female children | 0.333 | P<0.001 |
| Percentage of rural people | 0.098 | P<0.001 |
| Percentage of poor | 0.387 | P<0.001 |
| Percentage of Illiterate mother | 0.689 | P<0.001 |
| Percentage of children ever born to a women | 0.664 | P<0.001 |
| Percentage of Hindu people | 0.407 | P<0.001 |
| Percentage of thin mothers | 0.530 | P<0.001 |
| Percentage of children with low birth weight | 0.591 | P<0.001 |
| Percentage of children had diarrhoea | 0.357 | P<0.001 |

low MAD and MMF scores for stunting, underweight, and wasting depict a positive spatial autocorrelation. For MAD, Moran's I was 0.03 for stunting, 0.15 for underweight, and 0.16 for wasting. In the case of MMF, Moran's I values of 0.07 for stunting, 0.268 for underweight, and 0.34 for wasting depict a significant positive spatial autocorrelation. **Figs 5–7** depicts the spatial association between child feeding practices and child malnutrition in India. Data found a total of 47districts, especially the northern part of Uttar Pradesh, had the highest spatial clustering for a high prevalence of stunting against MMF (Fig 5). The bivariate LISA map for wasting and the clustering of child feeding practices shows that the central region of India had spatial clustering for MAD (87 districts), MMF (133 districts), and MDD (90) (Fig 6). Similarly, spatial clustering of 86 districts with a high prevalence of underweight children was observed in Uttar Pradesh and Bihar against a lower percentage of MMF (Fig 7). The clustering of low MDD and MAD with a higher prevalence of underweight could be observed across 34 and 58 districts, respectively.

The results of the Spatial Error Model demonstrate the significance of spatial autocorrelation of the correlates with stunting, underweight, and wasting across the districts of India (**Table 5**). After adjusting the socioeconomic characteristics, the result shows that not receiving MAD, MMF, and MDD was strongly associated with wasting and underweight. For MMF,

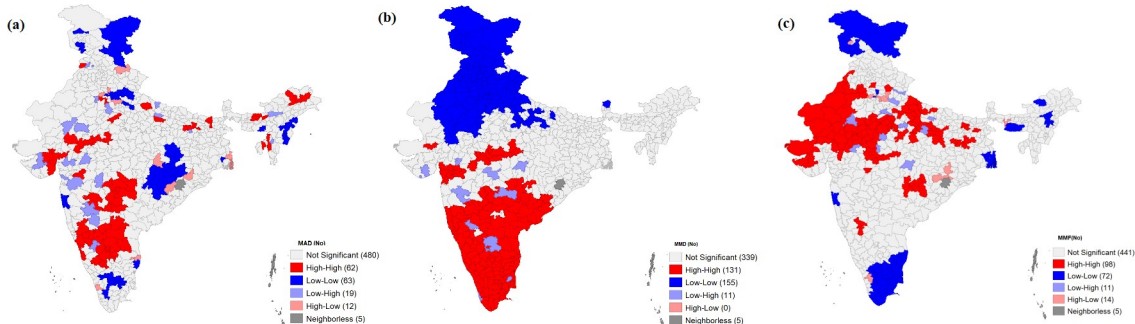

**Fig 4. Map showing the clustering of districts for MAD, MDD and MMF among children aged 6–23 months, India, 2015–16.**

**Table 4. Bivariate Local Moran's I results for all districts (n-640) among children aged 6–23 months, India, 2015–16.**

| Background Characteristics | Stunting | | Underweight | | Wasting | |
|---|---|---|---|---|---|---|
| | Moran's I | P-Value | Moran's I | P-Value | Moran's I | P-Value |
| Minimum Dietary Diversity (No) | -0.049 | P<0.004 | 0.016 | P<0.167 | 0.108 | P<0.001 |
| Minimum Acceptable Diet (No) | 0.038 | P<0.024 | 0.154 | P<0.001 | 0.161 | P<0.001 |
| Minimum Meal Frequency (No) | 0.074 | P<0.001 | 0.268 | P<0.001 | 0.347 | P<0.001 |
| Percentage of female children | 0.017 | P<0.197 | 0.244 | P<0.001 | 0.415 | P<0.001 |
| Percentage of rural people | 0.006 | P<0.337 | 0.119 | P< | 0.100 | P<0.001 |
| Percentage of poor people | 0.064 | P<0.001 | 0.218 | P<0.001 | 0.211 | P<0.001 |
| Percentage of Illiterate mother | 0.095 | P<0.001 | 0.367 | P<0.001 | 0.421 | P<0.001 |
| Percentage of children ever born to a women | 0.117 | P<0.001 | 0.386 | P<0.001 | 0.435 | P<0.001 |
| Percentage of Hindu people | 0.133 | P<0.001 | 0.312 | P<0.001 | 0.276 | P<0.001 |
| Percentage of thin mothers | 0.030 | P<0.056 | 0.174 | P<0.001 | 0.332 | P<0.001 |
| Percentage of children with low birth weight | 0.041 | P<0.014 | 0.353 | P<0.001 | 0.535 | P<0.001 |
| Percentage of children had diarrhoea | 0.088 | P<0.001 | 0.107 | P<0.001 | 0.211 | P<0.001 |

the coefficient was the largest for underweight (coef. = 0.113, z-value = 3.67) and wasting (coef. = 0.110, z-value = 5.54), indicating that not receiving the minimum frequency of diet was responsible for underweight and wasting. As for MDD, it had a strong positive relation with wasting (coef. = 0.011, z-value = 6.35). Chronic malnutrition or stunting was negatively associated with not receiving proper child feeding practices.

## Discussion

For the last few decades, the global supply and consumption of food have been consistently increasing [27]. Despite that, India ranks 114[th] out of 132 countries on under-five stunting and 120[th] out of 130 countries on under-five wasting [28]. The Indian consumption pattern is shifting from staple to non-staple foods [29]. Nevertheless, the supply of non-staple commodities such as pulses, vegetables, edible oils, dairy, meat, fruits, and fish lags compared to the demand [30]. Agriculture is not just about food production and income generation; it is also tied to nutrition [31]. Thus, sustainable agriculture has become a concern for policymakers as a weapon to reduce malnutrition in India. Records establish that the low productivity of crops affects food production, which is usually a significant cause of malnutrition [32]. Our study investigated the spatial association between child feeding practices and malnutrition on one side and crop yield across the districts of India on the other. We found that only 35%, 22%, and 9% of children aged 6–23 months received MMF, MDD, and MAD, respectively, in India.

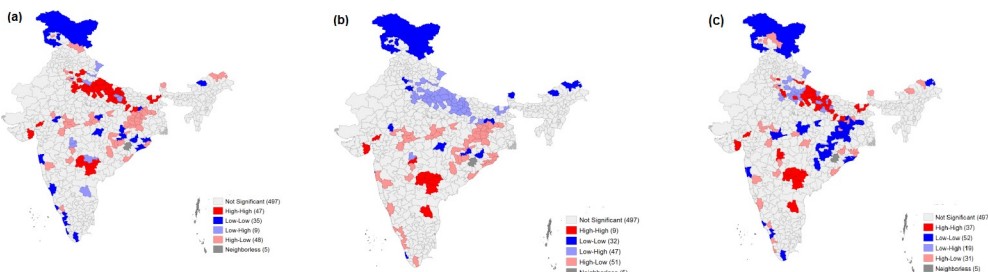

**Fig 5.** Bivariate LISA cluster maps showing the spatial clustering of stunting with a. Minimum meal frequency (MMF) b. Minimum dietary diversity (MDD) and c. Minimum acceptable diet (MAD) among children aged 6–23 months, India, 2015–16.

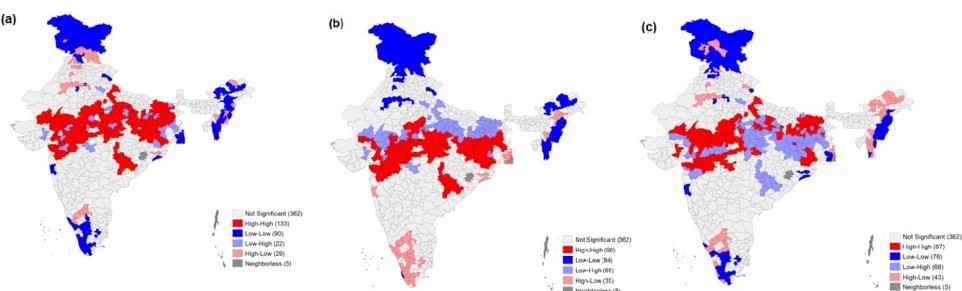

**Fig 6.** Bivariate LISA cluster maps showing the spatial clustering of wasting with a. Minimum meal frequency (MMF) b. Minimum dietary diversity (MDD) and c. Minimum acceptable diet (MAD) among children aged 6–23 months, India, 2015–16.

This is a significant factor behind wasting and underweight in India. The present study results are consistent with the findings of the past studies [33–35]. Chronic malnutrition or stunting is the result of inadequate consumption of micronutrients [36].

The study found that child feeding practices, such as MMF, MDD, and MAD, were positively associated with high yield rates of spices and cereals in India. The yield rate of cash crops, on the contrary, harmed child feeding practices. Production of pulses had a significant positive association on MAD and MDD. Districts with high cereal yield rates ensured that children receive MMF and MAD. The present study also demonstrated a significant spatial association between child feeding practices and malnutrition in India. A previous study from Myanmar remarked that agricultural production has a significant positive role in improving acute malnutrition [37]. Also, it has been found that agricultural intervention is a best practice to improve complementary feeding among children and, therefore, their nutritional status [38].

As expected, we found that not receiving a minimum diversity and frequency of meals and a minimum acceptable diet were significantly correlated with high yield rates of cereals. Agriculture, together with its allied sectors, is the largest source of livelihood in India. Fifty percent of the farmers depend on cereal crops in India. Most rural households(around 70%) still depend primarily on agriculture for their livelihood, with 82% of farmers being small and marginal [39]. It has been found that agricultural production affects household consumption expenditure, which ensures food security, especially among small farmers [40]. There is limited evidence on the relationship between crop yield rate and child nutrition. One study did find that cash crops are positively associated with income increment [41]. However, if the local

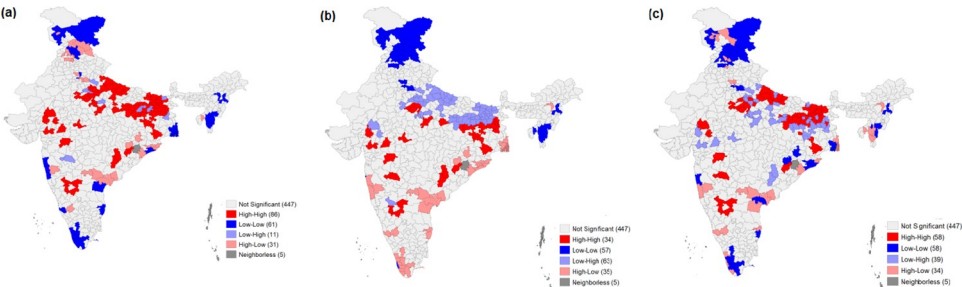

**Fig 7.** Bivariate LISA cluster maps showing the spatial clustering of underweight with a. Minimum meal frequency (MMF) b. Minimum dietary diversity (MDD) and c. Minimum acceptable diet (MAD) among children aged 6–23 months, India, 2015–16.

**Table 5. Estimated results from the Spatial Error Model for stunting, underweight and wasting in districts of India, 2015–16.**

| District level determinants | Stunting | | | Underweight | | | Wasting | | |
|---|---|---|---|---|---|---|---|---|---|
| | Beta | Z value | p value | Beta | Z value | p value | Beta | Z value | p value |
| Minimum Dietary Diversity (No) | -0.016 | -7.06 | p<0.001 | 0.008 | 0.92 | 0.356 | 0.011 | 6.35 | p<0.001 |
| Minimum Meal Frequency (No) | -0.069 | -3.63 | p<0.001 | 0.113 | 3.67 | p<0.001 | 0.110 | 5.54 | p<0.001 |
| Minimum Acceptable Diet (No) | 0.045 | 2.39 | p<0.016 | 0.020 | 4.40 | p<0.439 | -0.091 | -4.50 | p<0.001 |
| Percentage of female | 0.112 | 4.14 | p<0.001 | -0.108 | 0.77 | p<0.004 | 0.478 | 16.28 | p<0.001 |
| Percentage of rural population | 0.359 | 12.83 | p<0.001 | 0.002 | -2.85 | p<0.955 | -0.266 | -8.50 | p<0.001 |
| Percentage of Poor people | -0.059 | -4.45 | p<0.001 | 0.002 | 0.05 | p<0.876 | 0.028 | 1.99 | p<0.046 |
| Percentage of Illiterate female | 0.031 | 1.59 | p<0.110 | 0.005 | 0.15 | p<0.022 | 0.009 | 0.50 | p<0.615 |
| Percentage of children ever born to a women | -0.080 | -3.26 | p<0.001 | 0.105 | 2.28 | p<0.001 | 0.152 | 6.24 | p<0.001 |
| Percentage of Hindu | 0.109 | -4.71 | p<0.001 | 0.233 | 3.38 | p<0.001 | 0.142 | 6.01 | p<0.001 |
| Percentage of Thin female | -0.027 | -2.06 | p<0.038 | 0.081 | 7.67 | p<0.001 | 0.080 | 6.42 | p<0.001 |
| Percentage of children with low birth weight | 0.020 | 0.68 | p<0.494 | 0.015 | 5.11 | p<0.765 | 0.174 | 4.26 | p<0.001 |
| Percentage of children had diarrhoea | -0.012 | -2.72 | p<0.006 | 0.013 | 0.21 | p<0.828 | 0.053 | 1.10 | p<0.270 |
| Lambda Value (Lag coef.) | 0.698 | 20.65 | p<0.001 | 0.303 | 5.63 | p<0.001 | 0.355 | 6.84 | p<0.001 |
| AIC value | 3980 | | | 4406 | | | 4067 | | |
| Pseudo R Square | 0.73 | | | 0.59 | | | 0.81 | | |
| No of district | 640 | | | | | | | | |

markets fail to deliver the necessary quantities of food items, the outcome is somewhat different [42]. In India, the cropping pattern is shifting from food grain crops to high-profitable crops. As per historical records, in this transition stage, a negative relationship between cash crops and household food security can occur due to mainly lack of knowledge, infrastructure, and technology [43]. In India, about 10357 farmers commit suicide yearly [44], and the failure of cash crops plays a significant role in this[45]. Also with an increase in the proportion of cash crops, women's involvement in agriculture dips[46]. These factors, directly or indirectly, affect household income, local food availability, and women's involvement in food selection. Unsurprisingly, our study found a negative relationship between cash crops and child feeding practices, which indirectly affects the nutritional status of children and has been confirmed by a previous study [42].

Past studies have found strong linkages between agriculture and nutrition [47–49]. India is the largest consumer and producer of spices and pulses in the world [39]. Due to suitable soil and climatic conditions, south India is the home of spices. Spice farming is profitable due to the high worldwide demand for Indian spices. The area under cultivation and the production of spices have been showing a rising trend in India [50]. A previous study found that more than 50% of households of southern India consume at least 11 spices weekly [51]. Spices such as cumin, clove, garlic, bishop's weed, turmeric, and pepper benefit health [52]. Although the area under cultivation of spices is increasing, cereals are the major crop in India. However, both cereal and non-cereal (spices) farming can be done together. This is why a high diversity of food and non-food crops is observed in southern India [53], which may be another reason for the observed positive association between the yield rate of spices and better child feeding practices. Previous studies have demonstrated that sustainable agriculture is the most suitable approach to improve child feeding practices, which, in turn, determine child nutrition. Past studies also found that the local availability of food ensures household security, which is a sustainable approach to defeat malnutrition.

There are several strengths of this study. Firstly, this is the first study to reveal the linkage between agricultural productivity and child malnutrition through child feeding practices at the

district-level using large representative data. Secondly, the results are helpful for district-specific policies and programs and add to the existing evidence on determinants of child nutrition in India. However, the cross-sectional design of the NFHS-4 limits the causal inferences drawn from this analysis. Market factors and the nature of food distribution might affect child feeding practices but were not included in the analysis due to data lack of data. Moreover, the information on dietary habits was collected through 24 hours' recall data; hence there is a possibility of reporting/recall bias.

## Conclusions

Agricultural diversity at the district level is significantly associated with child feeding practices and thus contributes to the better nutritional status of children, though the socioeconomic status of the households moderates its influence. There is spatial clustering in the malnutrition status of the children. Additionally, our study revealed a spatial association between the child feeding practices and the nutritional status of children across the Indian districts. The study suggests that adopting nutrient-sensitive agriculture may be the best approach to improving children's nutritional status. Children tend to be fed healthy diets due to their local availability, although often parents may be unaware of the nutritional value of the foods they feed to their children. Encouraging local production of crops or food is thus expected to address malnutrition among children, helping the country attain SDG Goal 1 to alleviate malnutrition by 2030.

## Author Contributions

**Conceptualization:** Deepshikha Dey, Arup Jana, Manas Ranjan Pradhan.

**Data curation:** Deepshikha Dey, Arup Jana.

**Formal analysis:** Deepshikha Dey.

**Methodology:** Arup Jana.

**Software:** Deepshikha Dey.

**Supervision:** Manas Ranjan Pradhan.

**Writing – original draft:** Deepshikha Dey.

**Writing – review & editing:** Manas Ranjan Pradhan.

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
