## [Decision Letter · Decision Letter 0]

26 Jul 2021

PONE-D-21-18510

Influence of Agriculture on Child Nutrition through Child Feeding Practices in India: A District-Level Analysis

PLOS ONE

Dear Dr. Pradhan,

Thank you for submitting your manuscript to PLOS ONE. After careful consideration, we feel that it has merit but does not fully meet PLOS ONE’s publication criteria as it currently stands. Therefore, we invite you to submit a revised version of the manuscript that addresses the points raised during the review process.

 While the manuscript has contained good information, the needs a better organization, editing and data analysis and interpretation for addressing food availability and quality related to children health. 

The authors asked to address the comments of the reviewer(s) accordingly and thoroughly in a question and answer session. Also, address abbreviations and acronyms and in the abstract with data presentation in the table and figures with significant digits. There were 15 graphs which could be combined at 4 in 1. Important regression analyses can be shown in figures for better depiction of the results and interpretation.      

We look forward to receiving your revised manuscript.

Kind regards,

Rafiq Islam, Ph.D.

Academic Editor

PLOS ONE

Journal Requirements:

2.Please address the following:

- Please refrain from stating p values as 0.000 and instead use the format p<0.0001. 

- Please ensure you do not have any statements of causation within your Conclusions, as these cannot be confirmed following a study of this type

Additional Editor Comments (if provided):

While the manuscript has contained good information, the needs a better organization, editing and data analysis and interpretation for addressing food availability and quality related to children health. Also, address abbreviations and acronyms and in the abstract with data presentation in the table and figures with significant digits. There were 15 graphs which could be combined at 4 in 1. Important regression analyses can be shown in figures for better depiction of the results and interpretation.

Reviewers' comments:

Reviewer's Responses to Questions

**Comments to the Author**

1. Is the manuscript technically sound, and do the data support the conclusions?

Reviewer #1: Yes

2. Has the statistical analysis been performed appropriately and rigorously? 

Reviewer #1: Yes

3. Have the authors made all data underlying the findings in their manuscript fully available?

Reviewer #1: Yes

4. Is the manuscript presented in an intelligible fashion and written in standard English?

Reviewer #1: Yes

5. Review Comments to the Author

Reviewer #1: In this current research work, the authors have nicely presented influence of agriculture on child nutrition in India. However, there can be some improvements in the current manuscript. They are as follows-

1. Authors must add a conceptual or theoretical graphical framework in the introduction part of the present paper.

2. Materials and method section can be modified by defining the study districts, state, and the country more prominently under the study area section.

3. In analysis part, for deeper understanding and to find out stronger predictor variables the authors can go for stepwise regression, path, and principal component analysis (PCA). Additionally, authors must go for data visualization after the analyses.

4. In this present study, the total calorie consumed by the child per day in the study districts of India, can be added as an important independent variable.

5. BMI of both the mother and child can also be considered as one important dependent variable.

6. Some more references must be added in this current study.

6. PLOS authors have the option to publish the peer review history of their article (what does this mean?). If published, this will include your full peer review and any attached files.

Reviewer #1: **Yes: **Riti Chatterjee

---

## [Author Response · Author response to Decision Letter 0]

7 Sep 2021

Influence of Agriculture on Child Nutrition through Child Feeding Practices in India: A District-Level Analysis (PONE-D-21-18510)

(Research Article)

Manas Ranjan Pradhan (Corresponding Author)

International Institute for Population Sciences, India

Journal Requirements:

Comment 1: Please ensure that your manuscript meets PLOS ONE's style requirements, including those for file naming. 

Response 1: We have checked the PLOS ONE's style requirements, including those for file naming. 

Comment 2. Please address the following:

- Please refrain from stating p values as 0.000 and instead use the format p<0.0001. 

- Please ensure you do not have any statements of causation within your Conclusions, as these cannot be confirmed following a study of this type

Response 2: We have revised the manuscript as suggested

Additional Editor Comments (if provided):

Comment: While the manuscript has contained good information, the needs a better organization, editing and data analysis and interpretation for addressing food availability and quality related to children health. Also, address abbreviations and acronyms and in the abstract with data presentation in the table and figures with significant digits. There were 15 graphs which could be combined at 4 in 1. Important regression analyses can be shown in figures for better depiction of the results and interpretation.

Response: We have revised the paper in view of the comments. Specifically, addressed abbreviations and acronyms in the abstract, data presentation in the tables with significant digits, combined the graphs and presented the important regression results in figure.

Reviewer Comments

In this current research work, the authors have nicely presented influence of agriculture on child nutrition in India. However, there can be some improvements in the current manuscript. They are as follows-

Response: Thank you for reviewing our article and for the critical comments to improve the quality of the article. We have revised the article in view of the comments.

Comment 1: Authors must add a conceptual or theoretical graphical framework in the introduction part of the present paper.

Response 1: We have added a conceptual framework as per the suggestion.

Comment 2: Materials and method section can be modified by defining the study districts, state, and the country more prominently under the study area section.

Response 2: We have revised the materials and method section as suggested. Specifically, we have added a section on study area.

Comment 3: In analysis part, for deeper understanding and to find out stronger predictor variables the authors can go for stepwise regression, path, and principal component analysis (PCA). Additionally, authors must go for data visualization after the analyses.

Response 3: The authors have done data visualization as suggested. Specifically, the OLS regression result for Minimum Acceptable Diet (MAD) is presented through a forest plot. We restricted ourselves from making more such figures to limit the number of figures. The stepwise regression, path and PCA are alternative ways to look into the data though they do have disadvantages as well. We think OLS analysis is equally appropriate, given the objectives of the study and the nature of the data considered for this analysis.

Comment 4: In this present study, the total calorie consumed by the child per day in the study districts of India, can be added as an important independent variable.

Response 4: Thank you for the suggestion. However, the survey data considered for this analysis does not have this information. Nevertheless, we believe that MAD which is a composite indicator composed of MMF and MDD, indicate household calorie availability as well as consumption and hence can be considered as a proxy for calorie intake.

Comment 5: BMI of both the mother and child can also be considered as one important dependent variable.

Response 5: In this paper the authors aim is to assess the influence of agriculture on child undernutrition measured through stunting, wasting and underweight. To make the analysis focused, the authors did not include other associated outcome variables. Moreover, we have used the BMI of the mother as predictor variable in the analysis.

Comment 6: Some more references must be added in this current study.

Response 6: We have added additional references in the revised manuscript

---

## [Decision Letter · Decision Letter 1]

26 Nov 2021

Influence of Agriculture on Child Nutrition through Child Feeding Practices in India: A District-Level Analysis

PONE-D-21-18510R1

Dear Dr. Pradhan,

We’re pleased to inform you that your manuscript has been judged scientifically suitable for publication and will be formally accepted for publication once it meets all outstanding technical requirements.

Kind regards,

Rafiq Islam, Ph.D.

Academic Editor

PLOS ONE

Additional Editor Comments (optional):

Reviewers' comments:

Reviewer's Responses to Questions

**Comments to the Author**

1. If the authors have adequately addressed your comments raised in a previous round of review and you feel that this manuscript is now acceptable for publication, you may indicate that here to bypass the “Comments to the Author” section, enter your conflict of interest statement in the “Confidential to Editor” section, and submit your "Accept" recommendation.

Reviewer #1: All comments have been addressed

2. Is the manuscript technically sound, and do the data support the conclusions?

Reviewer #1: Yes

3. Has the statistical analysis been performed appropriately and rigorously? 

Reviewer #1: Yes

4. Have the authors made all data underlying the findings in their manuscript fully available?

Reviewer #1: Yes

5. Is the manuscript presented in an intelligible fashion and written in standard English?

Reviewer #1: Yes

6. Review Comments to the Author

Reviewer #1: In this current research work, the authors have nicely presented influence of agriculture on child nutrition in India.

After the first revision the authors nicely modified the points suggested by the reviewer. Now can be accepted.

7. PLOS authors have the option to publish the peer review history of their article (what does this mean?). If published, this will include your full peer review and any attached files.

Reviewer #1: **Yes: **Riti Chatterjee

---

## [Editor Report · Acceptance letter]

2 Dec 2021

PONE-D-21-18510R1 

Influence of Agriculture on Child Nutrition through Child Feeding Practices in India: A District-Level Analysis 

Dear Dr. Pradhan:

I'm pleased to inform you that your manuscript has been deemed suitable for publication in PLOS ONE. Congratulations! Your manuscript is now with our production department. 

Kind regards, 

on behalf of

Dr. Rafiq Islam 

Academic Editor

PLOS ONE